# Genetics of Hearing Impairment in North-Eastern Romania—A Cost-Effective Improved Diagnosis and Literature Review

**DOI:** 10.3390/genes11121506

**Published:** 2020-12-15

**Authors:** Irina Resmerita, Romica Sebastian Cozma, Roxana Popescu, Luminita Mihaela Radulescu, Monica Cristina Panzaru, Lacramioara Ionela Butnariu, Lavinia Caba, Ovidiu-Dumitru Ilie, Eva-Cristiana Gavril, Eusebiu Vlad Gorduza, Cristina Rusu

**Affiliations:** 1Department of Medical Genetics, Faculty of Medicine, “Grigore T. Popa” University of Medicine and Pharmacy, University Street, No 16, 700115 Iasi, Romania; roxana.popescu2014@gmail.com (R.P.); monica.panzaru@umfiasi.ro (M.C.P.); lacrybutnariu@gmail.com (L.I.B.); lavinia_zanet@yahoo.com (L.C.); evagavril@yahoo.com (E.-C.G.); vgord@mail.com (E.V.G.); abcrusu@gmail.com (C.R.); 2Department of Otorhinolaryngology, Faculty of Medicine, “Grigore T. Popa” University of Medicine and Pharmacy, University Street, No 16, 700115 Iasi, Romania; lmradulescu@yahoo.com; 3Department of Biology, Faculty of Biology, “Alexandru Ioan Cuza” University, Carol I Avenue, No 20A, 700505 Iasi, Romania; ovidiuilie90@yahoo.com

**Keywords:** hearing impairment, *GJB2*, NSHI, genetic screening, MLPA, cost-effective diagnosis

## Abstract

Background: We have investigated the main genetic causes for non-syndromic hearing impairment (NSHI) in the hearing impairment individuals from the North-Eastern Romania and proposed a cost-effective diagnosis protocol. Methods: MLPA followed by Sanger Sequencing were used for all 291 patients included in this study. Results: MLPA revealed abnormal results in 141 cases (48.45%): 57 (40.5%) were c.35delG homozygous, 26 (18.44%) were c.35delG heterozygous, 14 (9.93%) were compound heterozygous and 16 (11.35%) had other types of variants. The entire coding region of *GJB2* was sequenced and out of 150 patients with normal results at MLPA, 29.33% had abnormal results: variants in heterozygous state: c.71G>A (28%), c.457G>A (20%), c.269T>C (12%), c.109G>A (12%), c.100A>T (12%), c.551G>C (8%). Out of 26 patients with c.35delG in heterozygous state, 38.46% were in fact compound heterozygous. Conclusions: We identified two variants: c.109G>A and c.100A>T that have not been reported in any study from Romania. MLPA is an inexpensive, rapid and reliable technique that could be a cost-effective diagnosis method, useful for patients with hearing impairment. It can be adaptable for the mutation spectrum in every population and followed by Sanger sequencing can provide a genetic diagnosis for patients with different degrees of hearing impairment.

## 1. Introduction

Hearing impairment (HI) is the most common and heterogeneous sensory deficiency. It is defined by a unilateral or bilateral decrease in hearing acuity, more precisely a decrease in the hearing threshold in decibels (dB), at different frequencies. World Health Organization (WHO, Geneva, Switzerland) estimates that HI affects 466 million people around the world (6.1% of the world’s population), of which 34 million children. It is considered that 1/1000 newborns have a form of congenital hearing impairment [1,2]

More than 50% of cases of deafness are due to genetic causes [3] out of which 67% are classified as non-syndromic hearing impairment (NSHI) (no clinical findings that define a recognizable syndrome are associated), whereas a specific syndrome can be identified in 33% of cases [4].

In the last 5 years, progress has been made in identifying new hearing impairment genetic causes, due to research and new technology. Approximately 121 loci for NSHI have been currently mapped: 49 autosomal dominant, 76 autosomal recessive and 5 X-linked [5].

*GJB2* (NM_004004.5) or Gap Junction Protein β 2, situated on chromosome 13q12 (DFNB1 locus), is the most common cause of congenital hearing loss in many populations [6] including European and Mediterranean countries [7,8,9]. More than 150 different pathogenic variants in *GJB2* have been reported. The most frequent variant in the Caucasian populations is c.35delG, representing about 60% of all cases of NSHI [7,10,11,12].

*GJB3* (Gap Junction Protein β 3) and *GJB6* (Gap Junction Protein β 6) are the next frequent genes that can cause hearing impairment but they are less common, with less than 10 mutations cited [13,14,15].

The aim of this study was to identify and investigate the main genetic causes for NSHI in the hearing impairment subjects from the North-Eastern Romania and to convince other specialties to advice for genetic testing and counseling. Subsequently we verified a possibility to use Multiplex Ligation-dependent Probe Amplification (MLPA) as a cost-effective diagnosis protocol for developing countries and as a first intention genetic method. Genetic screening is feasible, *GJB2* being accountable for a large proportion of NSHI.

MLPA is a technique that can analyze in a single reaction up to 50 DNA sequences and detect copy number variations of several human genes, including small intragenic rearrangements but also single-nucleotide polymorphisms or aberrant DNA methylation. To our knowledge, the studies of hearing loss in Romania are based on RFLP (restriction fragment length polymorphism), ARMS-PCR (amplification refractory mutation system-polymerase chain reaction) analysis and Sanger Sequencing. Our Genetic Centre has experience in MLPA for other pathologies since 2012, so we tried to implement it as a screening method for hearing impairment individuals.

In patients with hearing impairment, the diagnostic approach starts with personal medical history, physical examination and family history for at least three generations and should continue with genetic tests and appropriate management.

## 2. Materials and Methods

### 2.1. Ethical Compliance

The patients included in the study were registered under a numerical code in order to maintain anonymity. The use of the results was done according to a protocol approved by the Ethics Commission of “Grigore T. Popa” University of Medicine and Pharmacy Iasi (approval No. 14789) and the Ethics Commission of “Saint Mary” Emergency Children’s Hospital Iasi (approval No. 681). Informed consent was signed by from patients, parents or legal guardians before beginning the research. All subjects included in this study were offered voluntary entrance.

### 2.2. Patient Recruitment

In the study (2015–2019), were enrolled 395 subjects with mild to profound and bilateral hearing impairment from the Iasi Regional Center for Medical Genetics and Audiology Department of Iasi Rehabilitation Clinical Hospital. All the subjects were clinically characterized by physical and auditory examinations. A number of 104 individuals were excluded from this study based on: syndromic or environmental/infectious etiology for hearing loss. The limitation of this study is that it investigates only the hearing loss subjects in order to assess the prevalence of certain type of mutations, so it does not include a control group.

### 2.3. Audiologic Assessment

Auditory functional assessment was performed only in the absence of the pathology of the middle ear, confirmed by otomicroscopy and wideband tympanometry. In cases identified with otitis media, the appropriate treatment was recommended and the child was rescheduled for repeated controls until the condition of the middle ear allowed audiological testing (normal otomicroscopy with wideband tympanogram of type A).

The audiological evaluation was adapted to the age and to the psycho-intellectual development of children. Thus, in children over 6 years of age, the auditory thresholds were measured by standard liminal tonal audiometry (air and bone conduction for 250 to 8000 Hz). In those under 6 years of age, as well as in some children over 6 years of age but who could not collaborate in subjective audiometric testing, the identification of hearing thresholds was done by objective audiological assessment of cross check type. The auditory steady state response and brainstem evoked response audiometry using insert headphones and distortion product otoacoustic emissions were measured in natural sleep. We performed visual reinforced audiometry or/and free field audiometric examination for subjective threshold confirmation in children who collaborated. In these cases, the conduction hearing loss was excluded based on normal otomicroscopy accompanied by type A wideband tympanometry. The audiologic evaluations were performed in soundproof rooms, using Interacoustics equipment (Equinox audiometer and Eclipse EP25). The audiologic follow-up was made periodically with the same methods adapted to each child’s particularity (age and medical condition), mainly at 4-, 6- or 12-months intervals, in order to identify the dynamic evolution of hearing impairment and for the fitting of the conventional hearing aids. Children with progressive hearing loss received the indication for cochlear implantation when they were included in the category of severe or profound hearing loss.

All subjects had cranial computed tomography (CT) scan and none showed ear malformation.

### 2.4. Research Methodology

a. DNA genomic extraction

DNA was extracted from 3 mL of peripheral blood samples stored with EDTA agent, using Wizard Genomic DNA Purification Kit (Promega Corp., Madison, WI, USA).

b. MLPA

The probe mix P163 *GJB-WFS1-POU3F4* was used for the detection of deletions or duplications in the *GJB2*, *GJB3*, *GJB6*, *POU3F4* genes, genomic microdeletions upstream of *POU3F4* and the presence of six specific variants in the *GJB2* gene: c.313_326del14, c.235delC, c.167delT, c.101T>C and c.35delG.

The MLPA analysis was performed according to the manufacturer’s protocol. Briefly, 100 nanograms of genomic DNA was denatured and hybridized with SALSA probes at 60 °C for approximately 17 h. PCR was performed after 15 min ligation at 54 °C, using Cy5 labeled primers. Fluorescent amplification products were separated based on their length by capillary electrophoresis in a CEQ 8000 GeXP Genetic Analysis System (Beckman Coulter, Brea, CA, USA) and the results were analyzed using Coffalyser.NET V9 program (MRC-Holland, Amsterdam, The Netherlands).

The probe ratio of deletion and duplication were fixed at 0.7 and 1.3 respectively.

Genomic regions of the *GJB2* gene were sequenced bidirectionally in heterozygous or normal individuals.

c. Sanger Sequencing

All samples were analyzed at the University of Medicine and Pharmacy “Grigore T. Popa” Iasi. The amplification using 125ng genomic DNA (25 µL reaction volume) was performed in a Sensoquest Thermocycler (Sensoquest, Göttingen, Germany), using Herculase II Fusion DNA Polymerase (Agilent Technologies, Santa Clara, CA, USA). PCR conditions included: initial denaturation (10 min at 95 °C), followed by 35 cycles of denaturation (30 s at 94 °C), annealing (30 s at 57 °C) and elongation (60 s at 72 °C), with a final elongation at 72 °C for 5 min, as described by M. RamShankar [16].

The sequencing was performed using primers previously described [16] and GenomeLab™ Dye Terminator Cycle Sequencing (DTCS) Quick Start kit (Beckman Coulter, Brea, CA, USA). A modified protocol was used with 10 µL reaction volume according to Azadan et al. [17]. The Agencourt system (Beckman-Coulter) was used to purify PCR amplicons (Agencourt AMPure XP, Brea, CA, USA) and sequencing products (Agencourt Cleanseq^®^ system, Brea, CA USA). The final products were subsequently separated and detected on a CEQ 8000 GeXP Genetic Analysis System (Beckman-Coulter). Sequences were analyzed in both directions (forward and reverse) and compared with the NCBI reference sequence NM_004004, using Mega6 software. The variants were verified for pathogenicity in Mutation taster, ClinVar and PolyPhen for the evaluation of disease-causing potential of sequence alterations [18,19,20].

d. Statistical Analysis

Experiment results were analyzed in Excel and presented in descriptive statistics.

## 3. Results

A total sample of 291 patients from North-Eastern Romania were collected between 2015–2019. HI was reported to be congenital and without other accompanying clinical features. All patients included in this study showed different pathologic levels of auditory thresholds, from mild to profound bilateral hearing impairment. The patients’ age ranged between 1 month to 52 years (median age 12.31).

Among the 291 probands, 74.6% (217/291) were sporadic cases of HI (simplex probands) (of which 15 with parental consanguinity) and 25.4% (74/291) had at least one first degree affected relative with bilateral HI (multiplex probands), of which 4 with parental consanguinity.

Mutations in *GJB2*, *GJB3*, *GJB6*, *POU3F4* and *WFS1* genes were analyzed by MLPA that revealed abnormal results in 141 cases (48.45%). Out of the total of 141 abnormal cases, 4 (2.84%) had variants in *WFS1* gene and 137 (97.16%) in *GJB2* gene: 57 (40.43%) were c.35delG homozygous, 26 (18.44%) were c.35delG heterozygous, 30 (21.28%) were compound heterozygous and 28 (19.86%) had other types of variants. No mutations were identified by MLPA in *GJB3*, *GJB6* and *POU3F4* genes.

Referring to the *WFS1* gene, all of the 4 patients with variants in this gene had exon 8 deletion (see Table 1). They had non-progressive mild to moderate hearing impairment and the age ranges from 15 to 20 years. We included these patients in a different study.

Regarding the *GJB2* gene, the most common pathogenic variant in the Romanian population is c.35delG, found in 97 patients in our study (33.3%). Of these, 57 patients (58.76%) had the c.35delG variant in homozygous state, 26 (26.84%) in heterozygous state and 14 (14.4%) were compound heterozygous for 3 different 35delG/non-35delG variants. 

The entire coding region of *GJB2* was sequenced in all individuals included in this study. Out of 26 patients with c.35delG variant in heterozygous state, 10 patients (38.46%) were in fact compound heterozygous. Among 150 patients with normal results at MLPA, 44 patients (29.33%) had abnormal results: 25 patients with variants in heterozygous state: 7 with c.71G>A (28%), 5 with c.457G>A (20%), 3 with c.269T>C (12%), 3 with c.109G>A (12%), 3 with c.100A>T (12%), 2 with c.551G>C (8%). All the patients with c.35delG in homozygous state were confirmed with Sanger Sequencing.

The *GJB2* variant spectrum found in this study is listed in Table 1.

Genotype-phenotype correlation was performed based on the distribution of the severity of HI in c.35delG and non-35delG genotype categories as shown in Table 2. Most cases had hearing loss before age 18. A small proportion of patients with mild hearing impairment showed a sequence variation in *GJB2*. Out of 18 patients with mild hearing loss, only 2 of them had c.35delG in homozygous state (diagnosed before age 4) and 4 patients had c.35delG in heterozygous state (diagnosed after the age 4). 81 patients had *GJB2* biallelic mutations and severe or profound hearing impairment: 20 (24.6%) of them had severe HI and c.35delG in homozygous state, 28 (34.5%) had profound HI and c.35delG in homozygous state, while 8 patients (9.87%) with severe HI had c.35delG in compound heterozygous state and 14 patients with profound HI had c.35delG in compound heterozygous state.

## 4. Discussion

Hearing impairment is one of the most heterogeneous conditions of considerable concern in medicine nowadays. Each population has a different etiologic profile based on ethnic, geographic, social and medical background. It is diagnosed in 1–2 of 1000 newborns [21], genetic factors are responsible to up to 2/3 of HI cases (70% non-syndromic and 30% syndromic deafness) [22]. The remaining one-third of cases can be caused by environmental and unidentified genetic factors.

The prevalence of *GJB2* gene mutations can vary according to ethnicity: more than 50% in the European population [23], 16% in China [24] and Iran [25], 9.6% in Mexican population [26] and 6.1% in Pakistan. Among the European population, the c.35delG variant represents 2/3 of the total mutations in the *GJB2* gene [27,28]. In other populations variants such as: c.235delC variant in Japanese and other Asian populations [29,30], c.167delT in the Ashkenazi Jews [31], c.71G>A in Indians and Roma [16,32] are prevalent.

Romania is a Latin country from Central-Eastern Europe and it is heterogenous from an ethnic point of view. The variant frequency and spectrum is different compared to other countries. The most important minorities in Romania are the Hungarian minority in North-West region, followed by Roma and other minorities.

We performed a genetic screening of *GJB2* gene (responsible for the major etiologies of hereditary hearing impairment among Romanians), *GJB3*, *GJB6, POU3F4* and *WFS1* genes. Genetic diagnosis was confirmed in 174 (59.7%) of the 291 patients with different degrees of hearing impairment, most of them being accounted for *GJB2* gene. This data is in accordance with the literature, *GJB2* mutations are frequent in all studied populations [13,14,33,34,35,36,37,38]. In some populations *GJB2* mutations are prevalent due to consanguineous marriages. In Turkey autosomal recessive inheritance is responsible for 76.9 % of the studied cases [39].

In this study we did not found any significant difference in the severity and evolution of hearing impairment when comparing the 74 multiplex probands with 217 simplex probands.

The c.35delG variant (rs80338939) is responsible for approximately 70% of autosomal recessive NSHI and is the most common cause of hearing loss in Caucasian populations [7,40]. The carrier rate is estimated to be the highest in Europe with a mean rate of 1.89% and a variation across countries with a higher rate of 2.48% in Southern Europe compared with 1.53% in Northern Europe [41]. This frequency was found also in hearing-impaired population from Hungary, Czech Republic, Poland and Austria, where c.35delG was prevalent [10,42,43,44]. In Romania there are relatively few data about the frequency and audiological features of *GJB2* gene sequence variants [45].

Because c.35delG is the most frequent variant in the coding region of the *GJB2* gene, it has become the first intention genetic investigation for patients with non-syndromic hearing loss.

In our study, the 35delG variant was present in 97 out of 291 (33.3%) patients with different degrees of hearing impairment. These results are in accordance with the findings previously reported in other Romanian and Central European studies [45,46]. All our patients with c.35delG variant were diagnosed by MLPA and confirmed with Sanger Sequencing of *GJB2* gene.

The study revealed that subjects with 35delG in homozygous state present more severe hearing impairment, compared with the 35delG/non-35delG compound heterozygotes. The subjects with two non-35delG variants have an even less degree of hearing impairment. This observation is in accordance with other studies which conclude that c.35delG in homozygous state is associated with a higher risk for severe hearing impairment [10,45,47,48,49].

The next frequent variant was c.101T>C (rs35887622), accounting for 19 out of 291 (6.5%) patients. In the Caucasian population the frequency of the c.101T>C variant was determined to be up to 6.5 % [50] and was initially reported as a polymorphism. Different studies of *GJB2* have determined that the c.101T>C variant is more frequent in individuals with mid-west American, UK [51] and German [50] origins, in comparison with those with French, Spanish, Italian and Japanese origin. A possible explanation may be that these variants are found in an ancestral mutation event that occurred in UK or Ireland. 

More than 50% of our patients with c.101T>C variant had moderate to profound hearing impairment. Also, at the time of the diagnosis, the age of the patients with c.101T>C was greater than the age of the patients with c.35delG. The results of a large study on the UK population affirmed that this variant is associated with mild/moderate HI [51]. The lower pathogenicity of the mutation that leads to later and milder manifestation of hearing impairment may sustain this finding. The majority of diagnosed cases with c.101T>C in our study were older than 18 years. One possible explanation can be that adults with mild forms of HI may not pursue audiology or genetic investigations. The progression of HI was found in few cases, because it was slow and long-term follow-up information was not possible.

The incomplete penetrance of c.101T>C variant was not confirmed because the study included only subjects with HI. The phenotype of the patients with hearing impairment was variable: the individuals with biallelic c.101T>C and c.35delG had moderate to profound hearing loss and the heterozygous c.101T>C had mild to moderate forms of hearing loss. No individuals with c.101T>C in homozygous state were found. 

The c.313_326del14 variant (rs111033253), called in the past c.310del14, c.312del14 or c.314del14, truncates the *GJB2* gene and disrupts the integrity of connexons. In many European populations, this variant has been identified previously with a frequency of pathogenic alleles from 0.47% to 28.3% [8].

The frequency of the c.313_326del14, variant in our group of participants was another finding in our study. The genotype was found in 8 of 291 patients (2.75 % of pathogenic alleles). A number of 6 patients were compound heterozygous with moderate to severe hearing impairment and the age of diagnosis being under 18 years and 2 patients had c.313-326del14 in heterozygous state with mild to moderate hearing impairment.

The c.71G>A variant is the fourth most common in our study and it had over five times lower frequency than the c.35delG variant. This mutation, previously called W24X, was first described in a Pakistani family [52] and later on was also discovered in several Asian families [53,54,55,56,57]. This finding indicates that it is the predominating cause of HI in India [58,59] and is prevalent in the Roma population with autosomal recessive NSHI [60]. In this study, it was found only in Roma patients: 15 individuals of 291 (15.5%) had this variant: 8 were compound heterozygous with c.35delG and 7 were in heterozygous state.

The c.71G>A frequency in different Roma subgroups is variable: it ranges from 0.0% to 26.1% in Slovak subgroups [60] and up to 4.0% in Spanish subgroups [61]. This finding is a result of the social structure of the Romani people, since they are considered to be a conglomerate of genetically isolated founder populations, with a high degree of consanguinity [61].

Our data is not concordant with other Central European series or the study from North-Western Romania, where 35delG and c.71G>A were the most common mutations [46], one explanation being the fact that the c.71G>A is predominant in Roma populations and they experience more problems accessing health care, from financial constraints, mobility issues or simply because they do not speak Romanian language.

The findings in Roma population confirm the ethnic origin of this mutation. Due to the fact that the sample of Roma patients is small, we cannot compare with other studies. The degree of addressability to medical care of the Roma-population from North-Eastern Romania is even lower than in North-Western Romania.

Another result determined in the present study was the presence of c.-23+1G>A, rs80338940 formerly called IVS1+1G>A, which is a splice site mutation found in exon 1 and intron 1 of *GJB2* gene in patients with hearing impairment. The mutation (revealed for the first time in 1999 by Denoyelle et al. [62]) was determined to be compound heterozygous and allele frequency was determined as 1% [63,64]. In our study, this variant was found in 4 of 291 patients (1.37%). The c.−23+1G>A variant was found with c.35delG variant and the subjects had severe to profound forms of HI. Previous studies showed that the patients with c.35delG/c.−23+1G>A in compound heterozygous state showed moderate HI [65] and profound HI [66]. To date, to our knowledge, homozygotes for the c.−23+1G>A variant have not been reported.

In 6 patients of 291 included in the study were identified two variants: c.109G>A and c.100A>T that have not been reported in any study from Romania. Out of these, 3 had c.109G>A variant and presented the same pattern of HI (progressive, bilateral and profound to severe) while the other 3 had c.100A>T (non-progressive, moderate, bilateral hearing impairment).

Our results contribute to define the mutation spectrum in the Romanian individuals with hearing impairment. Despite the genetic heterogeneity of NSHI, 217 patients were diagnosed out of a cohort of 291 patients. MLPA confirmed the genetic diagnosis in 141 cases (48.45%). We selected for further study the patients to which the *GJB2* mutations did not explain their hearing impairment and the patients with variants in *WFS1* gene.

Regarding our second aim of the study, we concluded that MLPA can be used as first intention genetic test for patients with HI due to some advantages over the Sanger Sequencing method: it is time saving, has a low price for consumables, the initial investment is lower for the platform than for Sanger Sequencing, the interpretation is much faster and it could easily detect the number copies variation and most frequent pathogenic variants. A description of the advantages and disadvantages is presented in Table 3.

This is the first report of the utility of MLPA and Sanger sequencing of HI in Romania; the results show notable findings in comparison to other European populations. However, some limitations should be noted: the samples included in this study are not truly representative for the entire Romania as all samples were collected from individuals with different degrees of hearing impairment, born in North-Eastern Romania, we did not have access to all of the parental samples to confirm compound heterozygosity. Our results need further studies on larger patient groups, especially Roma-population, in order to estimate the real incidence of the disease and to make more accurate predictions about the genotype phenotype correlation in our population.

We recommend genetic investigations in all subjects with hearing impairment that cannot be explained by other factors. In our knowledge, this is the first report on the utility and cost-effective of genetic testing in a cohort of Romanian patients with congenital NSHI. Sanger sequencing for *GJB2* gene is feasible because this is a small gene, with only 2 exons and the costs are reasonable and extra equipment are not necessary.

The genetic diagnosis in hearing impairment is important for many reasons: allows us to determine the etiology of deafness, offers the possibility to provide genetic counseling and prenatal diagnosis and not at least, based on the genotype-phenotype correlation provides prognostic information and facilitates an adequate management.

## 5. Conclusions

In this study, 217 patients had pathogenic/likely pathogenic variants, 141 being confirmed by MLPA. We identified two variants: c.109G>A and c.100A>T that have not being reported in any study from Romania. The most common variant in our study is c.35delG followed by c.101T>C, c.313_326del14 and c.71G>A.

All of the patients had been confirmed with Sanger Sequencing, proving that MLPA can be a cost-effective diagnosis method, useful for every patient with hearing impairment. MLPA is an inexpensive, rapid and reliable technique that could help as first intention genetic test for every individual with NSHI. Moreover, it can be adaptable for the mutation spectrum in every population and can be followed by Sanger sequencing for *GJB2* gene in cases of normal results.

## Figures and Tables

**Table 1 genes-11-01506-t001:** Variants spectrum found in this study.

	Variants	Protein Change	Clinical Significance	Patients (n)
MLPA	c.35delG, rs80338939	p.Gly12Valfs	Pathogenic	97
	c.101T>C, rs35887622	p.Met34Thr	Pathogenic	19
	c.313_326del14, rs111033253	p.Lys105Glyfs	Pathogenic	12
	c.-23+1G>A, rs80338940	p.Trp3Ter	Pathogenic	6
	Del *WFS* 1-8		Pathogenic	4
	Del ex1 *GJB2*		Pathogenic	3
SANGER SEQUENCING	c.71G>A, rs104894396	p.Trp24Ter	Pathogenic	15
	c.551G>C, rs80338950	p.Arg184Pro	Pathogenic	4
	c.109G>A, rs72474224	p.Val37Ile	Pathogenic	3
	c.269T>C, rs8033894	p.Leu90Pro	Pathogenic	3
	c.100A>T, rs564084861	p.Met34Leu	Uncertain significance	3
	c.457G>A, rs111033186	p.Val153Ile	Likely benign	5
	c.380G>A, rs111033196	p.Arg127His	Benign	10
	c.39G>A	p.(=)	Benign	4
	c.341C>G	p.Glu114Gly	Benign	4
	c.79G>A, rs2274084	p.Val27Ile	Benign	6

**Table 2 genes-11-01506-t002:** Correlations of *GJB2* genotypes and severity of hearing loss.

Genotypes		No of Subjects	Mild (21–40 dB)	Moderate (41–70 dB)	Severe (71–90 dB)	Profound (>90 dB)
**c.35delG Homozygous**	c.35delG/c.35delG	57	2	7	20	28
**c.35delG Heterozygous**	c.35delG/wt	26	4	11	7	4
**c.35delG Compound Heterozygous**	c.35delG/c.101T>C	10	-	6	2	2
	c.35delG/c.313_326del14	6	-	1	2	3
	c.35delG/c.-23+1G>A	4	-	-	1	3
	c.35delG/c.71G>A	8	-	1	3	4
	c.35delG/c.551G>C	2	-	-	-	2
**Non-35delG Compound Heterozygous**	c.79G>A/c.380G>A	1	-	1	-	-
	c.79G>A/c.341C>G/C.380G>A	4	1	3	-	-
	c.79G>A/c.39G>A	4	2	2	-	-
**Non-35delG Heterozygous**	c.101T>C/wt	9	5	2	1	1
	c.71G>A/wt	7	-	5	1	1
	c.457G>A/wt	5	1	2	2	-
	c.313_326del14/wt	2	-	1	1	-
	c.269T>C	3	-	-	2	1
	c.109G>A/wt	3	-	-	2	1
	c.551G>C/wt	2	-	1	1	-
	c.380G>A/wt	2	2	-	-	-
	c.100A>T	3	2	1	-	-
Total		158	19	44	45	50

**Table 3 genes-11-01506-t003:** Advantages and disadvantages for the methods used in this study [67,68].

Method	Advantages	Disadvantages
**MLPA**	Low costs	Sensitive to impurities
	Time efficient	Not suitable for unknown point mutations
	Free analysis software	
	High throughput	
	Can detect changes in the copy number, DNA methylation and known point mutation	
	Adaptable and updated	
**Sanger Sequencing**	Suitable for unknown point mutations	High costs
	Comprehensive coverage to any desired region	Time consuming
		Limited number of targets
		Sequence quality degrades after 700 to 900 bases

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
