# Peer review of "Genetics of Hearing Impairment in North-Eastern Romania—A Cost-Effective Improved Diagnosis and Literature Review"

_genes, 2020, doi:10.3390/genes11121506_

Round 1
Reviewer 1 Report
This manuscript provide a small insight in the prevalence of NSHI in a clinical sub-population of Romania. Despite the soundness of the aim there are several concerns:
Introduction : is well described but the aim should not be addressed as a general investigation of the genetic causes (it is impossibile without a control group in such small sample). It should be more orientated to define a preliminary exploration of the prevalence of genetic abnormalities in hearing loss subjects.
Please implement several limitations :
- poor audiometric testing (there is no exclusion of conductive hearing loss or middle ear diaseses)
- the sample is too small to assess any inference
- there is the lack of control group
Despite that the manuscript is well written and has a good style. It is very interesting the abnormalities found in the Roma sub groups and it worth for further studies.
Reviewer 2 Report
No comment
Author Response
No comment
Reviewer 3 Report
Genetics of hearing impairment in North-Eastern 2 Romania - a cost-effective improved diagnosis and literature review
This study by Resmerita et al. investigates the main genetic causes for NSHI in the North-Eastern region of Romania and proposes a cost-effective diagnosis protocol for developing countries. In this study, they used MLPA followed by Sanger Sequencing for 291 patients. They identified two variants: c.109G>A and c.100A>T that have not been reported in any study from Romania and conclude that MLPA is an inexpensive, rapid and reliable technique that could be a cost-effective diagnosis method, useful for every patient with hearing impairment. It can be adaptable for the mutation spectrum in every population and followed by Sanger sequencing can provide a genetic diagnosis for patients with different degrees of hearing impairment.
- More detailed information about how the previous literature has been done should be included in the introduction part. For example, is Multiplex Ligation-dependent Probe Amplification (MLPA) first used as a genetic test tool? If not, how has it been used in Romania or other countries? In line 63, this sentence looks weird here.
- In line 81, “Hearing levels were measured by pure tone audiometry, which included bone conduction.” However, the age range in this study is from 1 month old. So, the hearing levels were got several years later? More details should be given here.
- In line 135,” Mutations in GJB2, GJB3, GJB6, POU3F4, and WFS1 genes were analyzed by MLPA that revealed abnormal results in 141 cases (48.45%).” With MLPA, four types of genes were screened. However, only GJB2 and WFS1 genes mutation were described. How about the other three genes?
- In line 136,” 4 (2.84%) had variants in WFS1 gene”. I cannot find more information about these 4 WFS1 mutations. The authors need to indicate all the results from the genetic testing clearly.
- Some of the sentences are too long and hard to follow—for example, the sentence in line 168.
- As the second purpose of this study is to find a cost-effective diagnosis protocol. However, it lacks evidence that MPLA could be a more cost-effective protocol in gene screening. More information should be added in the discussion part. A table should be added to describe it.
- Some of the sentences should be reorganized. For example, in line 188,” GJB2 being responsible….”; in line 204, too many commas in the sentence. In line 233, “we did not include in our study normal hearing individuals.”
- In line 224-225, the author mentioned that the diagnosis age of c.101T>C was older than c.35delG.
- In line 278, the author stated that patients with c.109G>A had progressive bilateral and profound to severe hearing impairment. So, all these three patients showed the same pattern of hearing loss?
Round 2
Reviewer 1 Report
The authors have significantly improved the manuscript, and it is now acceptable for publication.